# A Comparison of Variational Bounds for the Information Bottleneck Functional

**DOI:** 10.3390/e22111229

**Published:** 2020-10-29

**Authors:** Bernhard C. Geiger, Ian S. Fischer

**Affiliations:** 1Know-Center GmbH, Inffeldgasse 13/6, 8010 Graz, Austria; 2Google Research, Mountain View, CA 94043, USA; iansf@google.com

**Keywords:** information bottleneck, deep learning, neural networks

## Abstract

In this short note, we relate the variational bounds proposed in Alemi et al. (2017) and Fischer (2020) for the information bottleneck (IB) and the conditional entropy bottleneck (CEB) functional, respectively. Although the two functionals were shown to be equivalent, it was empirically observed that optimizing bounds on the CEB functional achieves better generalization performance and adversarial robustness than optimizing those on the IB functional. This work tries to shed light on this issue by showing that, in the most general setting, no ordering can be established between these variational bounds, while such an ordering can be enforced by restricting the feasible sets over which the optimizations take place. The absence of such an ordering in the general setup suggests that the variational bound on the CEB functional is either more amenable to optimization or a relevant cost function for optimization in its own regard, i.e., without justification from the IB or CEB functionals.

## 1. Introduction

The celebrated information bottleneck (IB) functional [1] is a cost function for supervised lossy compression. More specifically, if *X* is an observation and *Y* a stochastically related random variable (RV) that we associate with relevance, then the IB problem aims to find an encoder $e_{Z|X}$, i.e., a conditional distribution of *Z* given *X*, that minimizes
(1)$$L_{\mathrm{IB}}:=I\left(X;Z\right)-\beta I\left(Y;Z\right).$$

In (Equation 1), I(X;Z) and I(Y;Z) denote the mutual information between observation *X* and representation *Z* and between relevant variable *Y* and representation *Z*, respectively, and β is a Lagrangian parameter. The aim is to obtain a representation *Z* that is simultaneously compressed (small I(X;Z)) and informative about the relevant variable *Y* (large I(Y;Z)), and the parameter β trades between these two goals.

Recently, Fischer proposed an equivalent formulation, termed the conditional entropy bottleneck (CEB) [2]. While the IB functional inherently assumes the Markov condition Y−X−Z, the CEB is motivated from the principle of Minimum Necessary Information, which lacks this Markov condition and which aims to find a representation *Z* that compresses a bi-variate dataset (X;Y) while still being useful for a given task. Instantiating the principle of Minimum Necessary Information induces then a Markov condition. For example, the task of finding a representation *Z* that makes *X* and *Y* conditionally independent induces the Markov condition X−Z−Y, and the representation optimal w.r.t. the principle of Minimum Necessary Information turns out to be $\arg\inf_{X-Z-Y}I\left(X,Y;Z\right)$, i.e., it is related to Wyner’s common information [3]. The task relevant in this work—estimating *Y* from a representation *Z* that is obtained exclusively from *X*—induces the Markov condition Y−X−Z and the constraint I(Y;Z)≥I(X;Y). A Lagrangian formulation of the constrained optimization problem $\inf_{I(Y;Z)\geq I(X;Y)}I\left(X;Z\right)$, where the infimum is taken over all encoders $e_{Z|X}$ that take only *X* as input, yields the CEB functional (see Section 2.3 of [2])
(2)$$L_{\mathrm{CEB}}:=I\left(X;Z|Y\right)-\gamma I\left(Y;Z\right).$$

Due to the chain rule of mutual information [4] (Theorem 2.5.2), (Equation 2) is equivalent to (Equation 1) for γ=β−1. Nevertheless, (Equation 2) has additional appeals. To this end, note that I(X;Z|Y) captures the information about *X* contained in the representation *Z* that is redundant for the task of predicting the class variable *Y*. In the language of [5], which essentially also proposed (Equation 2), I(X;Z|Y) thus quantifies class-conditional compression. Minimizing this class-conditional compression term I(X;Z|Y) is not in conflict with maximizing I(Y;Z), whereas minimizing I(X;Z) is (see Figure 2 in [2] and Section 2 in [5]). At the same time, as stated in [2] (p. 6), I(X;Z|Y) allows to “measure in absolute terms how much more we could compress our representation at the same predictive performance”, i.e., by how much I(X;Z|Y) could potentially be further reduced without simultaneously reducing I(Y;Z).

Aside from these theoretical considerations that make the CEB functional preferable over the equivalent IB functional, it has been shown that minimizing variational bounds on the former achieve better performance than minimizing variational bounds on the latter [2,6]. More specifically, it was shown that variational CEB (VCEB) achieves higher classification accuracy and better robustness against adversarial attacks than variational IB (VIB) proposed in [7].

The exact underlying reason why VCEB outperforms VIB is currently still being investigated. Comparing these two bounds at β−1=γ=1, Fischer suggests that “we may expect VIB to converge to a looser approximation of I(X;Z)=I(Y;Z)=I(X;Y)”, where the later equation corresponds to the Minimum Necessary Information point (see Section 2.5.1 of [2]). Furthermore, Fischer and Alemi claim that VCEB “can be thought of as a tighter variational approximation to the IB objective than VIB” (see Section 2.1 of [6]). Nevertheless, the following question remains: Does VCEB outperform VIB because the variational bound of VCEB is tighter, or because VCEB is more amenable to optimization than VIB?

To partly answer this question, we compare the optimization problems corresponding to VCEB and VIB. Rather than focusing on actual (commonly neural network-based) implementations of these problems, we keep an entirely mathematical perspective and discuss the problem of finding minimizers within well-defined feasible sets (see Section 3). Our main result in Section 4 shows that the optimization problems corresponding to VCEB and VIB are indeed ordered if additional constraints are added: If VCEB is constrained to use a consistent classifier-backward encoder pair (see Definition 1 below), then (unconstrained) VIB yields a tighter approximation of the IB functional. In contrast, if VIB is constrained to use a consistent classifier-marginal pair, then (constrained and unconstrained) VCEB yields a tighter approximation. If neither VCEB nor VIB are constrained, then no ordering can be shown between the resulting optimal variational bounds. Taken together, these results indicate that the superiority of VCEB over VIB observed in [2,6] cannot be due to VCEB better approximating the IB functional. Rather, we conclude in Section 5 that the variational bound provided in [2] is either more amenable to optimization, at least when the variational terms in VCEB and VIB are implemented using neural networks (NNs), or a successful cost function for optimization in its own regard, i.e., without justification from the IB or Minimum Necessary Information principles.

**Related Work and Scope.** Many variational bounds for mutual information have been proposed [8], and many of these bounds can be applied to the IB functional. Both the VIB and VCEB variational bounds belong to the class of Barber & Agakov bounds, cf. Section 2.1 of [8]. As an alternative example, the authors of [9] bounded the IB functional using the Donsker–Varadhan representation of mutual information. Aside from that, the IB functional has been used for NN training also without resorting to purely variational approaches. For example, the authors of [10] applied the Barber & Agakov bound to replace I(Y;Z) by the standard cross-entropy loss of a trained classifier, but used a non-parametric estimator for I(X;Z). Rather than comparing multiple variational bounds with each other, in this work we focus exclusively on the VIB [7] and VCEB [2] bounds. The structural similarity of these bounds allows a direct comparison and still yields interesting insights that can potentially carry over to other variational approaches.

We finally want to mention two works that draw conclusions similar to ours. First, Achille and Soatto [11] pointed to the fact that their choice of injecting multiplicative noise to neuron activations is not only a restriction of the feasible set over which the optimization is performed, but it can also be interpreted as a means of regularization or as an approach to perform optimization. Thus, the authors claim, there is an intricate connection between regularization (i.e., the cost function), the feasible set, and the method of optimization (see Section 9 of [11]); this claim resonates with our Section 5. Second, Wieczorek and Roth [12] investigate the difference between IB and VIB: While IB implicitly assumes the Markov condition Y−X−Z, the variational approach taken in VIB assumes that an estimate of *Y* is obtained from the representation *Z*, i.e., X−Z−Y. Dropping the former assumption allows to express the difference between the VIB bound and the IB functional via mutual and lautum information, which, taken together, measure the violation of the condition Y−X−Z. The authors thus argue that dropping this condition enables VIB and similar variants to optimize over larger sets of joint distributions of *X*, *Y*, and *Z*. In this work, we take a slightly different approach and argue that the posterior distribution of *Y* given *Z* is approximated by a classifier with input *Z* that responds with a class estimate $\hat{Y}$. Thus, we stick to the Markov condition inherent to IB and extend it by an additional variable, resulting in $Y-X-Z-\hat{Y}$. As a consequence, our variational approach does not assume that X−Z−Y holds, which also leads to a larger set of joint distributions of *X*, *Y*, and *Z*. Finally, while [12] compares the IB functional with the VIB bound, in our work we compare two variational bounds on the IB functional with each other.

**Notation.** We consider a classification task with a feature RV *X* on $R^{m}$ and a class RV *Y* on the finite set Y of classes. We assume that the joint distribution of *X* and *Y* is denoted by $p_{XY}$. In this work we are interested in representations *Z* of the feature RV *X*. This (typically real-valued) representation *Z* is obtained by feeding *X* to a stochastic encoder $e_{Z|X}$, and the representation *Z* can be used to infer the class label by feeding it to a classifier $c_{\hat{Y}|Z}$. Note that this classifier yields a class *estimate*
$\hat{Y}$ that need not coincide with the class RV *Y*. Thus, the setup of encoder, representation, and classifier yields the following Markov condition: $Y-X-Z-\hat{Y}$. We abuse notation and abbreviate the conditional probability (density) $p_{W|V=v}\left(\cdot \right)$ of a RV *W* given that another RV *V* assumes a certain value *v* as $p_{W|V}\left(\cdot |v\right)$. For example, the probability density of the representation *Z* for an input X=x is induced by the encoder $e_{Z|X}$ and is given as $e_{Z|X}\left(\cdot |x\right)$.

We obtain encoder, classifier, and eventual variational distributions via solving a constrained optimization problem. For example, $\min_{e_{Z|X}\in E}J$ minimizes the objective J over all encoders $e_{Z|X}$ from a given family E. In practice, encoder, classifier, and variational distributions are parameterized by (stochastic) feed-forward NNs. The chosen architecture has a certain influence on the feasible set; e.g., E may denote the set of encoders that can be parameterized by a NN of a given architecture.

We assume that the reader is familiar with information-theoretic quantities. More specifically, we let I(·;·) and $D\left(\cdot ∥\cdot \right)$ denote mutual information and Kullback–Leibler divergence, respectively. The expectation w.r.t. to a RV *W* drawn from a distribution $p_{W}$ is denoted as $E_{W∼p_{W}}\left[\cdot \right]$.

## 2. Variational Bounds on the Information Bottleneck Functional

We consider the IB principle for NN training. Specifically, we are interested in a (real-valued) representation *Z*, obtained directly from *X*, that minimizes the following functional:(3)$$L_{\mathrm{IB}}\left(\beta \right):=I\left(X;Z\right)-\beta I\left(Y;Z\right)=I\left(X;Z|Y\right)-\left(\beta -1\right)I\left(Y;Z\right)=:L_{\mathrm{CEB}}\left(\beta -1\right)$$

Rather than optimizing (Equation 3) directly (which was shown to be ill-advised at least for deterministic NNs in [13]), we rely on minimizing variational upper bounds. More specifically, the authors of [7] introduced the following variational bound on $L_{\mathrm{IB}}$:(4)$$L_{\mathrm{VIB}}\left(\beta \right):=E_{X∼p_{X}}\left[D\left(e_{Z|X}\left(\cdot |X\right)∥q_{Z}\right)\right]-\beta H\left(Y\right)-\beta E_{XYZ∼p_{XY}e_{Z|X}}\left[\log c_{\hat{Y}|Z}\left(Y|Z\right)\right]$$
where $e_{Z|X}$, $c_{\hat{Y}|Z}$, and $q_{Z}$ are called the encoder, classifier, and marginal. The classifier is used as a variational approximation to the distribution $p_{Y|Z}$. The marginal $q_{Z}$ is a learned distribution that aims to marginalize out the encoder $e_{Z|X}$. As such, this distribution is conceptually different from a fixed (unlearned) prior distribution in a Bayesian framework as in, e.g., the variational auto-encoder [14].

As an alternative and motivated by the principle of Minimum Necessary Information, the author of [2] proposed the variational bound on the CEB functional:(5)$$L_{\mathrm{VCEB}}\left(\beta \right):=E_{XY∼p_{XY}}\left[D\left(e_{Z|X}\left(\cdot |X\right)∥b_{Z|Y}\left(\cdot |Y\right)\right)\right]-\beta H\left(Y\right)-\beta E_{XYZ∼p_{XY}e_{Z|X}}\left[\log c_{\hat{Y}|Z}\left(Y|Z\right)\right]$$
where $b_{Z|Y}$ is called the backward encoder, which is a variational approximation to the distribution $p_{Z|Y}$.

## 3. Variational IB and Variational CEB as Optimization Problems

While it is known that $L_{\mathrm{IB}}\left(\beta \right)\leq L_{\mathrm{VIB}}\left(\beta \right)$ and $L_{\mathrm{IB}}\left(\beta \right)\leq L_{\mathrm{VCEB}}\left(\beta -1\right)$ for all possible $p_{XY}$ and all choices of $e_{Z|X}$, $b_{Z|Y}$, $c_{\hat{Y}|Z}$, and $q_{Z}$, it is not obvious how $L_{\mathrm{IB}}\left(\beta \right)$ and $L_{\mathrm{VCEB}}\left(\beta -1\right)$ compare during optimization. In other words, we are interested in determining whether there is an ordering between
(6a)$$\min_{e_{Z|X},c_{\hat{Y}|Z},q_{Z}}L_{\mathrm{VIB}}\left(\beta \right)$$
and
(6b)$$\min_{e_{Z|X},c_{\hat{Y}|Z},b_{Z|Y}}L_{\mathrm{VCEB}}\left(\beta -1\right).$$

Since we will always compare variational bounds for equivalent parameterization, i.e., compare $L_{\mathrm{VIB}}\left(\beta \right)$ with $L_{\mathrm{VCEB}}\left(\beta -1\right)$, we will drop the arguments β and β−1 for the sake of readability.

For a fair comparison, we need to ensure that both cost functions are optimized over comparable feasible sets E, C, B, and Q for the encoder, classifier, the backward encoder, and the marginal. We make this explicit in the following assumption.

**Assumption** **1.**
*The optimizations of VCEB and VIB are performed over equivalent feasible sets. Specifically, the families E and C from which VCEB and VIB can choose encoder $e_{Z|X}$ and classifier $c_{\hat{Y}|Z}$ shall be the same. Depending on the scenario, we may require that the optimization over the marginal $q_{Z}$ is able to choose from the same mixture models as are induced by VCEB. i.e., if $b_{Z|Y}\left(\cdot |y\right)$ is a feasible solution of $L_{\mathrm{VCEB}}$, then $q_{Z}\left(\cdot \right)=\sum_{y}b_{Z|Y}\left(\cdot |y\right)p_{Y}\left(y\right)$ shall also be a feasible solution for $L_{\mathrm{VIB}}$; we thus require that $Q⊇\{q_{Z}:\;q_{Z}\left(z\right)=\sum_{y}b_{Z|Y}\left(z|y\right)p_{Y}\left(y\right),\;b_{Z|Y}\in B\}$. Depending on the scenario, we may require that every feasible solution for the marginal $q_{Z}$ shall be achievable by selecting feasible backward encoders; we thus require that $B⊇\{b_{Z|Y}:\;q_{Z}\left(z\right)=\sum_{y}b_{Z|Y}\left(z|y\right)p_{Y}\left(y\right),\;q_{Z}\in Q\}$. If both conditions are fulfilled, then we write that B↔Q.*


We furthermore need the following definition:

**Definition** **1.**
*In the optimization of $L_{\mathrm{VCEB}}$, we say that backward encoder $b_{Z|Y}$ and classifier $c_{\hat{Y}|Z}$ are a consistent pair*
*if*
(7)$$c_{\hat{Y}|Z}\left(y|z\right)=\frac{p_{Y}\left(y\right)b_{Z|Y}\left(z|y\right)}{\sum_{y^{\prime }}p_{Y}\left(y^{\prime }\right)b_{Z|Y}\left(z|y^{\prime }\right)}=\frac{p_{Y}\left(y\right)b_{Z|Y}\left(z|y\right)}{q_{Z}^{\prime }\left(z\right)}$$
*holds. In the optimization of $L_{\mathrm{VIB}}$, we say that marginal $q_{Z}$ and classifier $c_{\hat{Y}|Z}$ are a consistent pair*
*if*
(8)$$p_{Y}\left(y\right)=\sum_{z}c_{\hat{Y}|Z}\left(y|z\right)q_{Z}\left(z\right)$$
*holds.*


The restriction to consistent pairs restricts the feasible sets. For example, for VCEB, if C is large enough to contain all classifiers consistent with backward encoders in B, i.e., if C
$⊇\{c_{\hat{Y}|Z}:\;c_{\hat{Y}|Z}\left(y|z\right)\propto p_{Y}\left(y\right)b_{Z|Y}\left(z|y\right),\;b_{Z|Y}\left(\cdot |y\right)\in B\}$, then the triple minimization
(9)$$\min_{\begin{array}{c} e_{Z|X}\in E,c_{\hat{Y}|Z}\in ,b_{Z|Y}\in B\\ (c_{\hat{Y}|Z},b_{Z|Y})\;\mathrm{consistent} \end{array}}L_{\mathrm{VCEB}}$$
is reduced to the double minimization
(10)$$\min_{e_{Z|X}\in E,b_{Z|Y}\in B}L_{\mathrm{VCEB}}.$$

Equivalently, one can write the joint triple minimization as a consecutive double minimization and a single minimization, where the inner minimization runs over all backwards encoders consistent with the classifier chosen in the outer minimization (where the minimization over an empty set returns infinity):(11)$$\min_{e_{Z|X}\in E,c_{\hat{Y}|Z}\in C}\left[\min_{b_{Z|Y}\in B\cap \left\{b_{Z|Y}^{\prime }:\;\frac{p_{Y}\left(y\right)b_{Z|Y}^{\prime }\left(z|y\right)}{\sum_{y^{\prime }}p_{Y}\left(y^{\prime }\right)b_{Z|Y}^{\prime }\left(z|y^{\prime }\right)}=c_{\hat{Y}|Z}\left(y|z\right)\right\}}L_{\mathrm{VCEB}}\right].$$

Similar considerations hold for VIB.

## 4. Main Results

Our first main result is negative in the sense that it shows $L_{\mathrm{VIB}}$ and $L_{\mathrm{VCEB}}$ cannot be ordered in general. To this end, consider the following two examples.

**Example** **1**(VIB < VCEB). *In this example, let B↔Q, where B and Q are constrained, and let C be unconstrained, thus $\min_{c_{\hat{Y}|Z}\in }-E_{XYZ∼p_{XY}e_{Z|X}}\left[\log c_{\hat{Y}|Z}\left(Y|Z\right)\right]=H\left(Y|Z\right)$. Suppose further that we have selected a fixed encoder $e_{Z|X}$ that induces the marginal and conditional distributions $p_{Z}$ and $p_{Z|Y}$, respectively. With this, we can write*
(12a)$$E_{XY∼p_{XY}}\left[D\left(e_{Z|X}\left(\cdot |X\right)∥b_{Z|Y}\left(\cdot |Y\right)\right)\right]=I\left(X;Z|Y\right)+E_{Y∼p_{Y}}\left[D\left(p_{Z|Y}\left(\cdot |Y\right)∥b_{Z|Y}\left(\cdot |Y\right)\right)\right]$$
*and*
(12b)$$E_{X∼p_{X}}\left[D\left(e_{Z|X}\left(\cdot |X\right)∥q_{Z}\right)\right]=I\left(X;Z\right)+D\left(p_{Z}∥q_{Z}\right).$$

*Suppose that $b_{Z|Y}^{\mathrm{VCEB}}$ is a minimizer of ([Disp-formula FD12a-entropy-22-01229]) over B and that $q_{Z}^{\mathrm{VCEB}}\left(z\right)=\sum_{y}p_{Y}\left(y\right)b_{Z|Y}^{\mathrm{VCEB}}\left(z|y\right)$. By the chain rule of of Kullback–Leibler divergence [4] (Th. 2.5.3) and with $b_{Y|Z}^{\mathrm{VCEB}}\left(y|z\right)=p_{Y}\left(y\right)b_{Z|Y}^{\mathrm{VCEB}}\left(z|y\right)/q_{Z}^{\mathrm{VCEB}}\left(z\right)$, we can expand*
$$\begin{array}{cc} D\left(p_{ZY}∥b_{Z|Y}^{\mathrm{VCEB}}p_{Y}\right)\; & =D\left(p_{Z}∥q_{Z}^{\mathrm{VCEB}}\right)+\underset{\geq 0}{\underset{︸}{E_{Z∼p_{Z}}\left[D\left(p_{Y|Z}\left(\cdot |Z\right)∥b_{Y|Z}^{\mathrm{VCEB}}\left(\cdot |Z\right)\right)\right]}}\\ \; & =\underset{=0}{\underset{︸}{D\left(p_{Y}∥p_{Y}\right)}}+E_{Y∼p_{Y}}\left[D\left(p_{Z|Y}\left(\cdot |Y\right)∥b_{Z|Y}^{\mathrm{VCEB}}\left(\cdot |Y\right)\right)\right] \end{array}$$
*thus*
$$E_{Y∼p_{Y}}\left[D\left(p_{Z|Y}\left(\cdot |Y\right)∥b_{Z|Y}^{\mathrm{VCEB}}\left(\cdot |Y\right)\right)\right]\geq D\left(p_{Z}∥q_{Z}^{\mathrm{VCEB}}\right).$$
*Suppose that $e_{Z|X}$ is such that the inequality above is strict. Then,*
$$\begin{array}{cc} \; & \min_{e_{Z|X}\in E,c_{\hat{Y}|Z}\in ,b_{Z|Y}\in B}L_{\mathrm{VCEB}}\left(\beta -1\right)\\ \; & =I\left(X;Z|Y\right)+E_{Y∼p_{Y}}\left[D\left(p_{Z|Y}\left(\cdot |Y\right)∥b_{Z|Y}^{\mathrm{VCEB}}\left(\cdot |Y\right)\right)\right]-\left(\beta -1\right)I\left(Y;Z\right)\\ \; & >I\left(X;Z|Y\right)+D\left(p_{Z}∥q_{Z}^{\mathrm{VCEB}}\right)-\left(\beta -1\right)I\left(Y;Z\right)\\ \; & =I\left(X;Z\right)+D\left(p_{Z}∥q_{Z}^{\mathrm{VCEB}}\right)-\beta I\left(Y;Z\right)\\ \; & \geq \min_{e_{Z|X}\in E,c_{\hat{Y}|Z}\in ,q_{Z}\in Q}L_{\mathrm{VIB}}\left(\beta \right) \end{array}$$

*where the last inequality follows because $q_{Z}^{\mathrm{VCEB}}$ may not be optimal for the VIB cost function.*


**Example** **2**(VIB > VCEB). *Let B↔Q, where Q and B are unconstrained, thus with (12) we have*
$$\begin{array}{c} \min_{b_{Z|Y}\in B}E_{XY∼p_{XY}}\left[D\left(e_{Z|X}\left(\cdot |X\right)∥b_{Z|Y}\left(\cdot |Y\right)\right)\right]=I\left(X;Z|Y\right)\\ \;and\;\min_{q_{Z}\in Q}E_{X∼p_{X}}\left[D\left(e_{Z|X}\left(\cdot |X\right)∥q_{Z}\right)\right]=I\left(X;Z\right). \end{array}$$
*Suppose further that C is such that $\min_{c_{\hat{Y}|Z}\in C}-E_{XYZ∼p_{XY}e_{Z|X}}\left[\log c_{\hat{Y}|Z}\left(Y|Z\right)\right]=H\left(Y|Z\right)+\varepsilon$, where ε>0. It then follows that*
$$\begin{array}{c} \min_{e_{Z|X}\in E,c_{\hat{Y}|Z}\in C,q_{Z}\in Q}L_{\mathrm{VIB}}\left(\beta \right)=I\left(X;Z\right)-\beta I\left(Y;Z\right)+\beta \varepsilon \\ >I\left(X;Z|Y\right)-\left(\beta -1\right)I\left(Y;Z\right)+\left(\beta -1\right)\varepsilon =\min_{e_{Z|X}\in E,c_{\hat{Y}|Z}\in C,b_{Z|Y}\in B}L_{\mathrm{VCEB}}\left(\beta -1\right). \end{array}$$


In both of these examples we have ensured that the comparison is fair in the sense of Assumption 1. Aside from showing that VIB and VCEB in general allow no ordering, additional interesting insights can be gleaned from Examples 1 and 2. First, whether VIB or VCEB yield tighter approximations of the IB and CEB functionals for a fixed encoder depends largely on the feasible sets C and B: Constraints on C cause disadvantages for VIB, while constraints on B lead to the VCEB bound becoming looser. Second, for fixed encoders, the tightness of the respective bounds and the question which of the bounds is tighter do not depend on how well the IB and CEB objectives are met: These objectives are functions only of the encoder $e_{Z|X}$, whereas the tightness of the variational bounds depends on C, B, and Q. (Of course, the tightness of the respective bounds after the triple optimization in (6) depends also on E, as the optimization over B and Q in Example 1 and over C in Example 2 interacts with the optimization over E in a non-trivial manner.)

Our second main result, in contrast, shows that the variational bounds can indeed be ordered if additional constraints are introduced. More specifically, if the variational bounds are restricted to consistent pairs as in Definition 1, then the following ordering can be shown. The proof of Theorem 1 is deferred to Section 6.

**Theorem** **1.**
*If VCEB is constrained to a consistent classifier-backward encoder pair, and if $Q⊇\{q_{Z}:\;q_{Z}\left(z\right)=\sum_{y}b_{Z|Y}\left(z|y\right)p_{Y}\left(y\right),\;b_{Z|Y}\in B\}$, then*
(13a)$$\min_{e_{Z|X}\in E,c_{\hat{Y}|Z}\in C,q_{Z}\in Q}L_{\mathrm{VIB}}\leq \min_{\begin{array}{c} e_{Z|X}\in E,c_{\hat{Y}|Z}\in C,b_{Z|Y}\in B\\ (c_{\hat{Y}|Z},b_{Z|Y})\;\mathrm{consistent} \end{array}}L_{\mathrm{VCEB}}.$$
*If VIB and VCEB are constrained to a consistent classifier–marginal and classifier-backward encoder pair, respectively, and if $B⊇\{b_{Z|Y}:\;b_{Z|Y}\left(z|y\right)=c_{\hat{Y}|Z}\left(y|z\right)q_{Z}\left(z\right)/p_{Y}\left(y\right),\;q_{Z}\in Q,c_{\hat{Y}|Z}\in C\}$, then*
(13b)$$\min_{\begin{array}{c} e_{Z|X}\in E,c_{\hat{Y}|Z}\in C,q_{Z}\in Q\\ (c_{\hat{Y}|Z},q_{Z})\;\mathrm{consistent} \end{array}}L_{\mathrm{VIB}}\geq \min_{\begin{array}{c} e_{Z|X}\in E,c_{\hat{Y}|Z}\in C,b_{Z|Y}\in B\\ (c_{\hat{Y}|Z},b_{Z|Y})\;\mathrm{consistent} \end{array}}L_{\mathrm{VCEB}}.$$
*A fortiori, ([Disp-formula FD13b-entropy-22-01229]) continues to hold if VCEB is not constrained to a consistent classifier-backward encoder pair.*


Theorem 1 thus relates the cost functions of VIB and VCEB in certain well-defined scenarios, contingent on the size of the feasible sets B and Q. If the variational approximations are implemented using NNs, then these bounds are thus contingent on the capacity of the NNs trained to represent the backward encoder in case of VCEB and the marginal in the case of VIB. A few clarifying statements are now in order.

First, it is easy to imagine scenarios in which the inequalities are strict. Trivially, this is the case for ([Disp-formula FD13a-entropy-22-01229]) if C and B, and for ([Disp-formula FD13b-entropy-22-01229]) if C and Q do not contain a consistent pair. Furthermore, if the set relations in the respective conditions do not hold with equality, the optimization over the strictly larger set of, e.g., marginals in ([Disp-formula FD13a-entropy-22-01229]), may yield strictly smaller values for the cost function $L_{\mathrm{VIB}}$.

Second, the condition that $B⊇\{b_{Z|Y}:\;b_{Z|Y}\left(z|y\right)=c_{\hat{Y}|Z}\left(y|z\right)q_{Z}\left(z\right)/p_{Y}\left(y\right),\;q_{Z}\in Q,c_{\hat{Y}|Z}\in C\}$ is less restrictive than the condition stated in Assumption 1. This is because every backward encoder that is written as $b_{Z|Y}\left(z|y\right)=c_{\hat{Y}|Z}^{\prime }\left(y|z\right)q_{Z}^{\prime }\left(z\right)/p_{Y}\left(y\right)$ for $q_{Z}^{\prime }\in Q$ and $c_{\hat{Y}|Z}^{\prime }\in C$ satisfies trivially that $\sum_{y}b_{Z|Y}^{\prime }\left(z|y\right)p_{Y}\left(y\right)=q_{Z}^{\prime }\left(z\right)$. Thus, if one accepts Assumption 1 as reasonable for a fair comparison between VCEB and VIB, then one must also accept that the ordering provided in the theorem is mainly a consequence of the restriction to consistent pairs, and not to one of the optimization problems having access to a significantly larger feasible set.

Finally, if C, B, and Q are sufficiently large, i.e., if the NNs implementing the classifier, backward encoder, and marginal are sufficiently powerful, then both VCEB and VIB can be assumed to yield equally good approximations of the IB functional. To see this, let $p_{Z}$, $p_{Z|Y}$, and $p_{Y|Z}$ denote the marginal and conditional distributions induced by $e_{Z|X}$ and note that with (12) we get
(14a)$$L_{\mathrm{VIB}}\left(\beta \right)=L_{\mathrm{IB}}\left(\beta \right)+D\left(p_{Z}∥q_{Z}\right)+\beta E_{Z∼p_{Z}}\left[D\left(p_{Y|Z}\left(\cdot |Z\right)∥c_{\hat{Y}|Z}\left(\cdot |Z\right)\right)\right]$$
and
(14b)$$\begin{array}{c} L_{\mathrm{VCEB}}\left(\beta -1\right)\\ =L_{\mathrm{IB}}\left(\beta \right)+E_{Y∼p_{Y}}\left[D\left(p_{Z|Y}\left(\cdot |Y\right)∥b_{Z|Y}\left(\cdot |Y\right)\right)\right]+\left(\beta -1\right)E_{Z∼p_{Z}}\left[D\left(p_{Y|Z}\left(\cdot |Z\right)∥c_{\hat{Y}|Z}\left(\cdot |Z\right)\right)\right]. \end{array}$$

Large B and Q render the second terms in both equations close to zero for all choices of $e_{Z|X}$ (see Example 2), while large C renders the last terms close to zero (see Example 1). Thus, in this case not only do we have $L_{\mathrm{VIB}}\left(\beta \right)\approx L_{\mathrm{VCEB}}\left(\beta -1\right)\approx L_{\mathrm{IB}}\left(\beta \right)$, but we also have that VCEB employs a consistent classifier-backward encoder pair by the fact that $b_{Z|Y}\approx p_{Z|Y}$ and $c_{\hat{Y}|Z}\approx p_{Y|Z}$. Thus, one may argue that if the feasible sets are sufficiently large, the restriction to consistent pairs may not lead to significantly looser bounds.

## 5. Discussion

In this note we have compared the IB and CEB functionals and their respective variational approximations. While IB and CEB are shown to be equivalent, the variational approximations VIB and VCEB yield different results after optimization. Specifically, it was observed that using VCEB as a training objective for stochastic NNs outperforms VIB in terms of accuracy, adversarial robustness, and out-of-distribution detection (see Section 3.1 of [2]). In our analysis we have observed that, although in general there is no ordering between VIB and VCEB (Examples 1 and 2), the optimal values of the cost functions can be ordered if additional restrictions are imposed (Theorem 1). Specifically, if VCEB is constrained to a consistent classifier-backward encoder pair, then its optimal value cannot fall below the optimal value of VIB. If, in contrast, VIB is constrained, then the optimal value of VIB cannot fall below the optimal value of VCEB (constrained or unconstrained). Thus, as expected, adding restrictions weakens the optimization problem w.r.t. the unconstrained counterpart.

These results imply that the superiority of VCEB is not caused by enabling a tighter bound on the IB functional than VIB does. Furthermore, it was shown in Table 1 of [6] that VCEB, constrained to a consistent classifier-backward encoder pair, yields better classification accuracy and robustness against corruptions than the unconstrained VCEB objective. Since obviously
(15)$$\min_{\begin{array}{c} e_{Z|X}\in E,c_{\hat{Y}|Z}\in ,b_{Z|Y}\in B\\ (c_{\hat{Y}|Z},b_{Z|Y})\;\mathrm{consistent} \end{array}}L_{\mathrm{VCEB}}\geq \min_{e_{Z|X}\in E,c_{\hat{Y}|Z}\in ,b_{Z|Y}\in B}L_{\mathrm{VCEB}}$$
the achievable tightness of a variational bound on the IB functional appears to be even negatively correlated with generalization performance in this set of experiments. (We note that [6] only reports constrained VCEB results for the largest NN models, and the constrained models perform slightly worse on robustness to adversarial examples than the unconstrained VCEB models of the same size.)

One may hypothesize, though, that VCEB is more amenable to optimization, in the sense that it achieves a tighter bound on the IB functional when encoder, classifier, and variational distributions are implemented and optimized using NNs. However, optimizing VCEB and VIB was shown to yield very similar results in terms of a lower bound on I(X;Z) for several values of β, cf. Figure 4 of [2], which seems not to support above hypothesis.

We therefore conclude that the superiority of (constrained) VCEB is not due to it better approximating the IB functional. While the hypothesis that the optimized VCEB functional approximates the optimized IB functional better cannot be ruled out, we will now formulate an alternative hypothesis. Namely, that the VCEB cost function itself instills desirable properties in the encoder that would otherwise not be instilled when relying exclusively on the IB functional, cf. Section 5.4 of [13]. For example, neither IB nor the Minimum Necessary Information principle include a classifier $c_{\hat{Y}|Z}$ in their formulations. Thus, by the invariance of mutual information under bijections, there may be many encoders $e_{Z|X}$ in the feasible set E that lead to representations *Z* equivalent in terms of (Equation 1) and (Equation 2). Only few of these representations are useful in the sense that the information about the class *Y* can be extracted “easily”. The variational approach of using a classifier to approximate I(Y;Z), however, ensures that, among all encoders $e_{Z|X}$ that are equivalent under the IB principle, one is chosen such that there exists a classifier $c_{\hat{Y}|Z}$ in C that allows inferring the class variable *Y* from *Z* with low entropy: While the IB and Minimum Necessary Information principles ensure that *Z* is informative about *Y*, the variational approaches of VIB and VCEB ensure that this information can be accessed in practice. Regarding the observed superiority of VCEB over VIB, one may argue that a variational bound relying on a backward encoder instills properties in the latent representation *Z* that are preferable over those that are achieved by optimizing a variational bound relying on a marginal only.

In other words, VCEB and VIB are justified as cost functions for NN training even without recourse to the IB and Minimum Necessary Information principles. This does not say that the concept of compression, inherent in both of these principles, is not a useful guidance—whether compression and generalization are causally related is the topic of an ongoing debate to which we do not want to contribute in this work. Rather, we claim that variational approaches may yield desirable properties that go beyond compression and that may be overlooked when too much focus is put on the functionals that are approximated with these variational bounds.

In combination with the variational approach, the selection of feasible sets can also have profound impact on the properties of the representation *Z*. A representation *Z* is called disentangled if its distribution $p_{Z}$ factorizes. Disentanglement can thus be measured by total correlation, i.e., the Kullback–Leibler divergence between $p_{Z}$ and the product of its marginals Section 5 of the [11]. Achille and Soatto have shown that selecting Q in the optimization of VIB as a family of factorized marginals is equivalent to adding a total correlation term to the IB functional, effectively encouraging disentanglement, cf. Proposition 1 in [11]. Similarly, Amjad and Geiger note that selecting B in the optimization of VCEB as a family of factorized backward encoders encourages class-conditional disentanglement; i.e., it enforces a Naive Bayes structure on the representation *Z*, cf. Corollary 1 & Section 3.1 of [5]. To understand the implications of these observations, it is important to note that neither disentanglement nor class-conditional disentanglement are encouraged by the IB or CEB functionals. However, by appropriately selecting the feasible sets of VIB or VCEB, disentanglement and class-conditional disentanglement can be achieved. While we leave it to the discretion of the reader to decide whether disentanglement is desirable or not, we believe that it is vital to understand that disentanglement is an achievement of optimizing a variational bound over an appropriately selected feasible set, and not one of the principles based on which these variational approaches are motivated.

## 6. Proof of Theorem 1

We start with the first assertion. Assume that $e_{Z|X}^{\mathrm{VCEB}}$, $b_{Z|Y}^{\mathrm{VCEB}}$, and $c_{\hat{Y}|Z}^{\mathrm{VCEB}}$ are the optimal encoder, backward encoder, and classifier in terms of the VCEB cost function under the assumption of consistency, i.e.,
(16)$$\begin{array}{c} \min_{\begin{array}{c} e_{Z|X}\in E,c_{\hat{Y}|Z}\in C,b_{Z|Y}\in B\\ (c_{\hat{Y}|Z},b_{Z|Y})\;\mathrm{consistent} \end{array}}L_{\mathrm{VCEB}}=E_{XY∼p_{XY}}\left[D\left(e_{Z|X}^{\mathrm{VCEB}}\left(\cdot |X\right)∥b_{Z|Y}^{\mathrm{VCEB}}\left(\cdot |Y\right)\right)\right]\\ -\left(\beta -1\right)H\left(Y\right)-\left(\beta -1\right)E_{XYZ∼p_{XY}e_{Z|X}^{\mathrm{VCEB}}}\left[\log c_{\hat{Y}|Z}^{\mathrm{VCEB}}\left(Y|Z\right)\right] \end{array}$$
where
(17)$$c_{\hat{Y}|Z}^{\mathrm{VCEB}}\left(y|z\right)=\frac{p_{Y}\left(y\right)b_{Z|Y}^{\mathrm{VCEB}}\left(z|y\right)}{\sum_{y^{\prime }}p_{Y}\left(y^{\prime }\right)b_{Z|Y}^{\mathrm{VCEB}}\left(z|y^{\prime }\right)}=\frac{p_{Y}\left(y\right)b_{Z|Y}^{\mathrm{VCEB}}\left(z|y\right)}{q_{Z}^{\prime }\left(z\right)}.$$

Certainly, if C and B are such that they do not admit a consistent pair, then this minimum is infinity and the inequality holds trivially.

For the VIB optimization problem, we obtain
$$\begin{array}{cc} \; & \min_{e_{Z|X}\in E,c_{\hat{Y}|Z}\in C,q_{Z}\in Q}L_{\mathrm{VIB}}\\ \; & =\min_{e_{Z|X}\in E,c_{\hat{Y}|Z}\in C,q_{Z}\in Q}E_{X∼p_{X}}\left[D\left(e_{Z|X}\left(\cdot |X\right)∥q_{Z}\right)\right]-\beta H\left(Y\right)-\beta E_{XYZ∼p_{XY}e_{Z|X}}\left[\log c_{\hat{Y}|Z}\left(Y|Z\right)\right]\\ \; & \overset{\left(a\right)}{=}\min_{e_{Z|X}\in E,c_{\hat{Y}|Z}\in C,q_{Z}\in Q}E_{XZ∼p_{X}e_{Z|X}}\left[\log\frac{e_{Z|X}\left(Z|X\right)}{q_{Z}\left(Z\right)}\right]-\beta H\left(Y\right)-\beta E_{XYZ∼p_{XY}e_{Z|X}}\left[\log c_{\hat{Y}|Z}\left(Y|Z\right)\right]\\ \; & \overset{\left(b\right)}{=}\min_{e_{Z|X}\in E,c_{\hat{Y}|Z}\in C,q_{Z}\in Q}E_{XYZ∼p_{XY}e_{Z|X}}\left[\log\frac{e_{Z|X}\left(\cdot |X\right)c_{\hat{Y}|Z}^{\mathrm{VCEB}}\left(Y|Z\right)}{q_{Z}\left(Z\right)c_{\hat{Y}|Z}^{\mathrm{VCEB}}\left(Y|Z\right)}\right]-\beta H\left(Y\right)\\ \; & \;\;-\beta E_{XYZ∼p_{XY}e_{Z|X}}\left[\log c_{\hat{Y}|Z}\left(Y|Z\right)\right]\\ \; & \overset{\left(c\right)}{\leq }\min_{e_{Z|X}\in E,q_{Z}\in Q}E_{XYZ∼p_{XY}e_{Z|X}}\left[\log\frac{e_{Z|X}\left(\cdot |X\right)c_{\hat{Y}|Z}^{\mathrm{VCEB}}\left(Y|Z\right)}{q_{Z}\left(Z\right)c_{\hat{Y}|Z}^{\mathrm{VCEB}}\left(Y|Z\right)}\right]-\beta H\left(Y\right)-\beta E_{XYZ∼p_{XY}e_{Z|X}}\left[\log c_{\hat{Y}|Z}^{\mathrm{VCEB}}\left(Y|Z\right)\right]\\ \; & \overset{\left(d\right)}{\leq }\min_{e_{Z|X}\in E}E_{XYZ∼p_{XY}e_{Z|X}}\left[\log\frac{e_{Z|X}\left(Z|X\right)c_{\hat{Y}|Z}^{\mathrm{VCEB}}\left(Y|Z\right)}{p_{Y}\left(Y\right)b_{Z|Y}^{\mathrm{VCEB}}\left(Z|Y\right)}\right]-\beta H\left(Y\right)-\beta E_{XYZ∼p_{XY}e_{Z|X}}\left[\log c_{\hat{Y}|Z}^{\mathrm{VCEB}}\left(Y|Z\right)\right]\\ \; & \overset{\left(e\right)}{=}\min_{e_{Z|X}\in E}E_{XY∼p_{XY}}\left[D\left(e_{Z|X}\left(\cdot |X\right)∥b_{Z|Y}^{\mathrm{VCEB}}\left(\cdot |Y\right)\right)\right]+E_{XYZ∼p_{XY}e_{Z|X}}\left[\log\frac{c_{\hat{Y}|Z}^{\mathrm{VCEB}}\left(Y|Z\right)}{p_{Y}\left(Y\right)}\right]\\ \; & \;\;-\beta H\left(Y\right)-\beta E_{XYZ∼p_{XY}e_{Z|X}}\left[\log c_{\hat{Y}|Z}^{\mathrm{VCEB}}\left(Y|Z\right)\right]\\ \; & \overset{\left(f\right)}{=}\min_{e_{Z|X}\in E}E_{XY∼p_{XY}}\left[D\left(e_{Z|X}\left(\cdot |X\right)∥b_{Z|Y}^{\mathrm{VCEB}}\left(\cdot |Y\right)\right)\right]-\left(\beta -1\right)H\left(Y\right)\\ \; & \;\;-\left(\beta -1\right)E_{XYZ∼p_{XY}e_{Z|X}}\left[\log c_{\hat{Y}|Z}^{\mathrm{VCEB}}\left(Y|Z\right)\right]\\ \; & \overset{\left(g\right)}{\leq }E_{XY∼p_{XY}}\left[D\left(e_{Z|X}^{\mathrm{VCEB}}\left(\cdot |X\right)∥b_{Z|Y}^{\mathrm{VCEB}}\left(\cdot |Y\right)\right)\right]-\left(\beta -1\right)H\left(Y\right)-\left(\beta -1\right)E_{XYZ∼p_{XY}e_{Z|X}^{\mathrm{VCEB}}}\left[\log c_{\hat{Y}|Z}^{\mathrm{VCEB}}\left(Y|Z\right)\right]\\ \; & =\min_{e_{Z|X}\in E,c_{\hat{Y}|Z}\in C,b_{Z|Y}\in B}L_{\mathrm{CEB}} \end{array}$$
where

(a) follows by writing the KL divergence as an expectation of the logarithm of a ratio;(b) follows by multiplying both numerator and denominator in the first term with $c_{\hat{Y}|Z}^{\mathrm{VCEB}}$;(c) is because of the (potential) suboptimality of $c_{\hat{Y}|Z}^{\mathrm{VCEB}}$ for the VIB cost function;(d) is because $Q⊇\{q_{Z}:\;q_{Z}\left(z\right)=\sum_{y}b_{Z|Y}\left(z|y\right)p_{Y}\left(y\right),\;b_{Z|Y}\in B\}$, thus we may choose $q_{Z}=q_{Z}^{\prime }$ where $q_{Z}^{\prime }$ is defined in (Equation 17); and because this particular choice may be suboptimal for the VIB cost function;(e) follows by splitting the logarithm(f) follows by noticing that $E_{XYZ∼p_{XY}e_{Z|X}}\left[\log p_{Y}\left(Y\right)\right]=-H\left(Y\right)$(g) follows because $e_{Z|X}^{\mathrm{VCEB}}$ may be suboptimal for the VIB cost function.

Comparing the last line with (Equation 16) completes the proof of the first assertion.

We next turn to the second assertion. Assume that $e_{Z|X}^{\mathrm{VIB}}$, $c_{\hat{Y}|Z}^{\mathrm{VIB}}$, and $q_{Z}^{\mathrm{VIB}}$ are the optimal encoder, classifier, and marginal in terms of the VIB cost function under the assumption of consistency, i.e.,
(18)$$\min_{\begin{array}{c} e_{Z|X}\in E,c_{\hat{Y}|Z}\in C,q_{Z}\in Q\\ (c_{\hat{Y}|Z},q_{Z})\;\mathrm{consistent} \end{array}}L_{\mathrm{VIB}}:=E_{X∼p_{X}}\left[D\left(e_{Z|X}^{\mathrm{VIB}}\left(\cdot |X\right)∥q_{Z}^{\mathrm{VIB}}\right)\right]-\beta H\left(Y\right)-\beta E_{XYZ∼p_{XY}e_{Z|X}^{\mathrm{VIB}}}\left[\log c_{\hat{Y}|Z}^{\mathrm{VIB}}\left(Y|Z\right)\right]$$
where
(19)$$p_{Y}\left(y\right)=\sum_{z}c_{\hat{Y}|Z}^{\mathrm{VIB}}\left(y|z\right)q_{Z}^{\mathrm{VIB}}\left(z\right).$$

Again, if C and Q are such that they do not admit a consistent pair, then this minimum is infinity and the inequality holds trivially.

For the VCEB optimization problem, we obtain
$$\begin{array}{cc} \; & \min_{\begin{array}{c} e_{Z|X}\in E,c_{\hat{Y}|Z}\in C,b_{Z|Y}\in B\\ (c_{\hat{Y}|Z},b_{Z|Y})\;\mathrm{consistent} \end{array}}L_{\mathrm{VCEB}}+\left(\beta -1\right)H\left(Y\right)\\ \; & =\min_{\begin{array}{c} e_{Z|X}\in E,c_{\hat{Y}|Z}\in C,b_{Z|Y}\in B\\ (c_{\hat{Y}|Z},b_{Z|Y})\;\mathrm{consistent} \end{array}}E_{XY∼p_{XY}}\left[D\left(e_{Z|X}\left(\cdot |X\right)∥b_{Z|Y}\left(\cdot |Y\right)\right)\right]-\left(\beta -1\right)E_{XYZ∼p_{XY}e_{Z|X}}\left[\log c_{\hat{Y}|Z}\left(Y|Z\right)\right]\\ \; & \overset{\left(a\right)}{=}\min_{\begin{array}{c} e_{Z|X}\in E,c_{\hat{Y}|Z}\in C,b_{Z|Y}\in B\\ (c_{\hat{Y}|Z},b_{Z|Y})\;\mathrm{consistent} \end{array}}E_{XYZ∼p_{XY}e_{Z|X}}\left[\log\frac{e_{Z|X}\left(Z|X\right)}{b_{Z|Y}\left(Y|Z\right)}\right]-\left(\beta -1\right)E_{XYZ∼p_{XY}e_{Z|X}}\left[\log c_{\hat{Y}|Z}\left(Y|Z\right)\right]\\ \; & \overset{\left(b\right)}{=}\min_{e_{Z|X}\in E,c_{\hat{Y}|Z}\in C}\min_{b_{Z|Y}\in B\cap \left\{b_{Z|Y}^{\prime }:\;\frac{p_{Y}\left(y\right)b_{Z|Y}^{\prime }\left(z|y\right)}{\sum_{y^{\prime }}p_{Y}\left(y^{\prime }\right)b_{Z|Y}^{\prime }\left(z|y^{\prime }\right)}=c_{\hat{Y}|Z}\left(y|z\right)\right\}}E_{XYZ∼p_{XY}e_{Z|X}}\left[\log\frac{e_{Z|X}\left(Z|X\right)}{b_{Z|Y}\left(Y|Z\right)}\right]\\ \; & \;\;-\left(\beta -1\right)E_{XYZ∼p_{XY}e_{Z|X}}\left[\log c_{\hat{Y}|Z}\left(Y|Z\right)\right]\\ \; & \overset{\left(c\right)}{\leq }\min_{e_{Z|X}\in E}\min_{b_{Z|Y}\in B\cap \left\{b_{Z|Y}^{\prime }:\;\frac{p_{Y}\left(y\right)b_{Z|Y}^{\prime }\left(z|y\right)}{\sum_{y^{\prime }}p_{Y}\left(y^{\prime }\right)b_{Z|Y}^{\prime }\left(z|y^{\prime }\right)}=c_{\hat{Y}|Z}^{\mathrm{VIB}}\left(y|z\right)\right\}}E_{XYZ∼p_{XY}e_{Z|X}}\left[\log\frac{e_{Z|X}\left(Z|X\right)}{b_{Z|Y}\left(Z|Y\right)}\right]\\ \; & \;\;-\left(\beta -1\right)E_{XYZ∼p_{XY}e_{Z|X}}\left[\log c_{\hat{Y}|Z}^{\mathrm{VIB}}\left(Y|Z\right)\right]\\ \; & \overset{\left(d\right)}{\leq }\min_{e_{Z|X}\in E}E_{XYZ∼p_{XY}e_{Z|X}}\left[\log\frac{e_{Z|X}\left(Z|X\right)}{c_{\hat{Y}|Z}^{\mathrm{VIB}}\left(Y|Z\right)q_{Z}^{\mathrm{VIB}}\left(Z\right)}\right]-H\left(Y\right)-\left(\beta -1\right)E_{XYZ∼p_{XY}e_{Z|X}}\left[\log c_{\hat{Y}|Z}^{\mathrm{VIB}}\left(Y|Z\right)\right]\\ \; & =\min_{e_{Z|X}\in E}E_{XZ∼p_{X}e_{Z|X}}\left[\log\frac{e_{Z|X}\left(Z|X\right)}{q_{Z}^{\mathrm{VIB}}\left(Z\right)}\right]-H\left(Y\right)-\beta E_{XYZ∼p_{XY}e_{Z|X}}\left[\log c_{\hat{Y}|Z}^{\mathrm{VIB}}\left(Y|Z\right)\right]\\ \; & \overset{\left(e\right)}{\leq }E_{XZ∼p_{X}e_{Z|X}^{\mathrm{VIB}}}\left[\log\frac{e_{Z|X}^{\mathrm{VIB}}\left(Z|X\right)}{q_{Z}^{\mathrm{VIB}}\left(Z\right)}\right]-H\left(Y\right)-\beta E_{XYZ∼p_{XY}e_{Z|X}^{\mathrm{VIB}}}\left[\log c_{\hat{Y}|Z}^{\mathrm{VIB}}\left(Y|Z\right)\right]\\ \; & =\min_{\begin{array}{c} e_{Z|X}\in E,c_{\hat{Y}|Z}\in C,q_{Z}\in Q\\ (c_{\hat{Y}|Z},q_{Z})\;\mathrm{consistent} \end{array}}L_{\mathrm{VIB}}+\left(\beta -1\right)H\left(Y\right) \end{array}$$
where

(a) follows by writing the KL divergence as an expectation of the logarithm of a ratio;(b) follows by the assumption that the VCEB problem is constrained to a consistent classifier-backward encoder pair, and from (Equation 11);(c) is because of the (potential) suboptimality of $c_{\hat{Y}|Z}^{\mathrm{VIB}}$ for the VCEB cost function;(d) follows by adding and subtracting H(Y); by choosing $b_{Z|Y}^{\mathrm{VIB}}=c_{\hat{Y}|Z}^{\mathrm{VIB}}q_{Z}^{\mathrm{VIB}}/p_{Y}$, which is possible because $B⊇\{b_{Z|Y}:\;b_{Z|Y}\left(z|y\right)=c_{\hat{Y}|Z}\left(y|z\right)q_{Z}\left(z\right)/p_{Y}\left(y\right),\;q_{Z}\in Q,c_{\hat{Y}|Z}\in C\}$; and by the fact that this particular choice may be suboptimal for the VCEB cost function;(e) follows because $e_{Z|X}^{\mathrm{VIB}}$ may be suboptimal for the VCEB cost function.

This completes the proof.□

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
