# Peer review of "A Comparison of Variational Bounds for the Information Bottleneck Functional"

_entropy, 2020, doi:10.3390/e22111229_

Round 1

Reviewer 1 Report

This paper considers the well-known Information Bottleneck model. Considering the Markov chain Y---X---Z, the goal of the (supervised) Information Bottleneck model is to obtain a representation Z from the observed data X, through the channel P_{Z \mid X}, wherein this scenario Y denotes a hidden parameter correlated with X, e.g., in facial images Y can denote an attribute of image X.   In the previous paper by Ian Fischer entitled "The conditional Entropy Bottleneck", the author addressed an alternative objective that instead of the complexity component I(X; Z), the conditional mutual information I(X; Z \mid Y) is considered. Noting that I (X; Z) = I (Y; Z) + I( X; Z \mid Y), indeed the authors have taken into an account the residual information in their objective. Clearly, the associated Lagrangian functional of both IB and CEB are equivalent.   This paper addresses the possible ordering between the variational bound in these two problems by restricting the feasible sets of encoder, classifier, backward encoder and the latent space prior. The first part of results shows that, as expected, there is no ordering between the variational bounds of VIB and VCEB, in general. The second part results show that if additional constraints are introduced, the variational bounds can be ordered.   The objective of this paper is clear and straightforward. I have a couple of comments to improve the paper:   1-In the third paragraph: the authors state, " ... the CEB motivated from the principle of Minimum ... " I am familiar with the Ian Ficher's paper on CEB. However, the statement about the absence of the Markov chain in the case of CEB /starting from line 17 "... which lacks this Markov condition and which aims to find a representation Z ..." should be better explained for the reader. The practical construction of the encoder that jointly compressed Y and X into Z should be well examplified. In fact, the reader is somehow confused with the assumed encoder in equation (5), i.e., $e_{Z \mid X}$.

  2-In line 19 and 20 (after Eq.~2): the authors said, " I ( X; Z \mid Y) captures the additional information about X contained in the representation ..."   This sentence is not precise and somehow misleading. When you say "additional information", the reader assumes that the CEB functional considers an extra component that is not captured in the IB model. However, one knows that it is just the residual information. Considering the IB and CEB scenarios, I would to say: " I ( X; Z \mid Y) captures the REDUNDANT information ...".   3- In section 2, q_Z is the " marginal (learned aggregated posterior) " or the "proposal prior"? I think it is more properly to refer to q_Z as a prior distribution. Please, clarify it.

Author Response

Thank you for your comments and for your positive evaluation of our manuscript! We appreciate the time and effort you have spent for providing this review. Below we will respond to each comment individually:

1) We agree that the MNI principle and its lack of a Markov condition are a bit vague at this point. To make matters more clear, we have added a second example, complementing the example of predicting Y from Z, where Z is obtained from X (Y-X-Z). Namely, we have added the task of finding a Z that makes X and Y conditionally independent. Then, the MNI is identical to the formulation of Wyner's common information. Also, we have made explicit that in the first example of predicting Y, the encoder takes only X as input.
2) We have changed the sentence to "To this end, note that I(X;Z|Y) captures the information about X contained in the representation Z that is redundant for the task of predicting the class variable Y."
3) We try to distinguish between learned distributions in a variational framework and prior distributions in Bayesian modeling. q_Z in this work falls into the first category, while the corresponding distribution in a variational auto-encoder falls into the second category. To make this clear and to justify our terminology, we have added the following sentence after the introduction of q_Z:

"The marginal q_Z is a learned distribution that aims to marginalize out the encoder e_Z|X. As such, this distribution is conceptually different from a fixed (unlearned) prior distribution in a Bayesian framework as in, e.g., the variational auto-encoder [cite]."

Reviewer 2 Report

Some comments are helpful for improving the work. 1. In Abstract, “we relate the variational bounds proposed in [1] and [2]”, the citing of references should be improved. No reference is suggested in Abstract for the independence of Abstract. 2. Definition 1 needs proof and/or the corresponding properties should be discussed. 3. The information is not complete for Reference 1.

Author Response

Thank you for your comments! We appreciate the time and effort you have spent for providing this review. Below we will respond to each comment individually:

1) We agree that the abstract should be self-contained and, thus, should usually not rely on references. In our case, a certain extent of referencing is indispensable. As the information for authors did not mention such cases, we would like to leave the decision how to display these citations to the editorial staff at Entropy.
2) We are not sure how to interpret this comment. There is no proof available for Definition 1, but we have tried our best to make its implications as clear as possible (see text between Definition 1 and Section 4). If you could give us more concrete pointers to where our explanations are unclear, we would be happy to improve the respective passages.
3) Indeed, several of our references appear incomplete (months and addresses are missing). This appears to be a problem with the LaTeX template, as our external bib-files are complete and contain this information. We hope that the editorial staff at Entropy can help us out in this regard.